# The Serum Levels of the Heavy Metals Cu, Zn, Cd, and Pb and Progression of COPD—A Preliminary Study

**DOI:** 10.3390/ijerph20021427

**Published:** 2023-01-12

**Authors:** Elica Valkova, Vasil Atanasov, Tatyana Vlaykova, Tanya Tacheva, Yanitsa Zhelyazkova, Dimo Dimov, Kristian Yakimov

**Affiliations:** 1Department of Biological Sciences, Agriculture Faculty, Trakia University, 6000 Stara Zagora, Bulgaria; 2Department of Medical Chemistry and Biochemistry, Medical Faculty, Trakia University, 6000 Stara Zagora, Bulgaria; 3Department of Medical Biochemistry, Medical University of Plovdiv, 4002 Plovdiv, Bulgaria

**Keywords:** Cu, Zn, Cd, Pb, markers for COPD, environmental pollution

## Abstract

There is evidence in previous studies that high levels of heavy metals may play a key role in the development of COPD due to the induction of chronic inflammation and oxidative stress. In this preliminary study, we used atomic absorption spectrophotometry to measure the levels of four heavy metals (Cu, Zn, Cd, and Pb) in blood serum of COPD patients and controls over 2 years. Clinical data on disease progression or absence were collected in patients living in the industrial region of Stara Zagora, Bulgaria. The mean values of Cu in the serum of patients with COPD and the control group were 374.29 ± 15.03 μg/L and 238.55 ± 175.31 μg/L, Zn—2010.435 ± 670.006 μg/L and 1672.78 ± 934.27 μg/L, Cd—0.334 ± 0.0216 μg/L and 0.395 ± 0.110 μg/L and Pb—0.0732 ± 0.009 μg/L and 0.075 ± 0.0153 μg/L. This is probably because these elements are biogenic and are used in the body for its anti-oxidant protection. In fact, it cannot be stated with certainty that elevated levels of Cu and Zn in the environment have a negative impact in COPD patients. There was a trend towards higher levels of the toxicants lead and cadmium in COPD patients compared to the control group of patients. There is a statistically unproven trend toward higher levels of lead and cadmium in COPD patients compared to controls, which to some extent supports our hypothesis that there is a relationship between environmental lead and cadmium levels and the COPD manifested. In COPD patients, a positive correlation was found between BMI and serum Cu levels (r = 0.413, *p* = 0.005). A higher concentration of serum Cu was found in men with BMI ≥ 30, compared to those with BMI < 30. There is also a positive correlation to a lesser extent between CRP and cadmium (r = 0.380; *p* = 0.019) and lead (r = 0.452; *p* = 0.004). The correlation of lead and cadmium with PSA also shows that these elements may also be associated with the presence of inflammatory processes. A significant negative correlation exists between Pb in the serum of patients with COPD and their blood hemoglobin (r = −356; *p* = 0.028). The results of our study suggest that higher doses of the trace elements Cu and Zn do not always have a negative effect in patients with COPD, while the toxicants Pb and Cd may be involved in COPD exacerbation and can be used as prognostic biomarkers for progression. Further studies are warranted to confirm these preliminary results.

## 1. Introduction

Chronic obstructive pulmonary disease (COPD) is characterized by inflammation and chronic persistent limitation of airflow. More people die each year from COPD than from lung cancer and breast cancer combined [1]. The airflow limitation is a result of remodeling of the airways and lung parenchyma, hyperplasia of airway epithelial cells, thickening of the basement membrane, deposition of collagen, peri-bronchial fibrosis and bronchial smooth muscle cell hyperplasia, resulting in progressive obstruction of the airways that is not fully reversible, and interferes with normal breathing [2].

The mechanistic basis underlying COPD is complex and can involve recurrent inflammation, oxidative stress and imbalance of protease/antiprotease [3]. Risk factors for the onset and development of COPD include a combination of diverse behavioral, environmental, and genetic components e.g., exposure to biomass smoke, occupational exposures to dust and fumes, outdoor air pollution [4]. However, smoking remains the most important cause of COPD in western countries [5]

Smoking can directly lead to the production of reactive oxygen species (ROS) or induce a variety of cellular responses (activation of macrophages and neutrophils) resulting in oxidative stress and inflammation. The systematic and local oxidative stress may alter remodeling of extracellular matrix (ECM) and blood vessels, stimulate mucus secretion, cause apoptosis and regulate cell proliferation and immune modulation [6,7].

There is evidence that high levels of heavy metals may play a key role in the development of COPD due to the induction of chronic inflammation and oxidative stress [8].

The presence of higher levels of heavy metals in the air, water and food that people consume on a daily basis is largely complemented by the effects of smoking. The region where the settlement subjected to research is located is also significant. The Stara Zagora region in Bulgaria is one of the high-risk regions in terms of environmental pollution, due to the presence of three large coal-fired thermal power plants in the Maritsa-Iztok complex nearby [9].

Not an insignificant percentage of the amounts of lead (Pb) and cadmium (Cd) in the body are due to the intake of these metals via the inhaled air [10]. The increase in serum Cd levels has been found to be directly related to decreased lung function in smokers [11,12].

Lead is characterized by multilateral toxicity. This metal has low chemical mobility and large distribution. It exists in the form of metallic lead, inorganic ions and salts [13]. It is contained in airborne dust particles, but enters the aquatic environment mainly through rainfall, mining of ore, production of batteries, chemicals and dyes [14]. Lead not only does not play a significant role from a biochemical aspect, but has a powerful toxic effect on all organisms. Toxic action of Pb manifests as acceleration of lysis of erythrocytes and damage to the brain, leading to tissue hypoxia and disorders in nervous system function [15,16,17].

Cadmium is characterized by a normal presence in the environment as an element of soil, air, sediments and even unpolluted sea and fresh water. It occurs mainly in the form of inorganic salts [18]. Global studies have found that Cd induces oxidative stress, leading to increased lipid peroxidation and the production of reactive oxygen species (ROS), resulting in tissue damage [19,20]. One of the main sources of cadmium exposure in humans is tobacco smoke. Because Cd absorption by the lungs is much greater than from the gastrointestinal tract, smoking contributes significantly to the overall workload of the body [11,21,22]. Cadmium has the ability to bind to the mitochondria and inhibit both cellular respiration and oxidative phosphorylation even in low concentrations [23].

Lead and cadmium suppress a number of processes in the body such as apoptosis, cell cycle regulation, DNA repair, DNA methylation, cell growth and differentiation [24]. These activities are directly related to the mechanism of DNA repair, the generation of ROS and the induction of apoptosis [25,26].

Zinc (Zn) plays a key role in many processes, such as DNA and RNA synthesis, energy metabolism, many metabolic reactions and in the regulation of the immune system. Increased prevalence of obstructive pulmonary disease has been associated with low daily dietary Zn intake [15].

Copper (Cu) is considered an essential element for living cells. On the other hand, it might be quite toxic, as the toxicity of Cu is directly dependent on the values of the physicochemical parameters of water (pH, alkalinity, solutes, hardness, etc.) [27,28].

Diagnosing COPD in the earliest stages is crucial. Despite presenting with respiratory symptoms or other disease manifestations, COPD patients are frequently diagnosed late. Late diagnosis has been associated with a higher risk of exacerbation and increased rate of hospitalizations [29]. In this respect, the detection of potential biomarkers of exposure could contribute to early COPD onset diagnosis and predicting the clinical outcomes of the disease.

Due to the fact that Stara Zagora is considered a polluted area, our hypothesis is that there is a relationship between environmental pollution with heavy metals in this region and the risk of COPD developing [29]. In this regard, the aim of our study was to determine the levels of heavy metals Pb, Cd, Cu and Zn in the blood serum of COPD patients and control group and to evaluate their possible role as biomarkers for COPD.

## 2. Materials and Methods

### 2.1. Patients and Controls

In the present study a total of 76 participants from the region of Stara Zagora were included (patients with COPD and control group without symptoms from the same ethnic group and area of Bulgaria). The patients were recruited at the Clinic of Internal Medicine, University Hospital, Trakia University, Stara Zagora, Bulgaria.

Patients had different stages of the disease according to GOLD (GOLD II, III and IV). The inclusion criteria for COPD were as the following: age higher than 40 years; forced expiratory volume in 1 s (FEV_1_) of <80%; forced expiratory volume in 1 s (FEV_1_)/forced vital capacity (FVC) ratio of ≤70%; FEV1 reversibility after inhalation of Salbutamol 400 µg of <12%.

The available demographic and clinical data are presented in Table 1. The ethics committee at Medical Faculty, Trakia University, Stara Zagora, Bulgaria approved the study protocol and informed consent was obtained from all study participants before the study.

### 2.2. Collection of Blood Serum Samples

Two ml of peripheral blood from the patients and controls was collected, kept at 4 °C for 30 min and centrifuged. The serum samples and whole blood were stored at −20 °C until the assays.

### 2.3. Determination of Quantities of Heavy Metals Cd, Pb, Cu and Zn

Serum sample preparation was performed by wet mineralization in a Perkin Elmer 3000 microwave oven. Each sample was mineralized in a mixture of 6 mL of concentrated nitric acid (65% HNO_3_) and 1 mL of concentrated hydrochloric acid (37%HCl), and an oxidizing agent—1 mL of hydrogen peroxide (30% H_2_O_2_) was added to clarify the solution. All reagents used for the analyses were chemically pure and produced by the Merck company. The content of the heavy elements Pb, Cd, Cu and Zn in the resulting acid solutions was determined by atomic absorption spectrometer (AAS) “A Analyst 800” Perkin Elmer on a cuvette and flame system by using acetylene–oxygen combustion according to ISO 11,047 [30].

### 2.4. Statistical Analyses

Statistical analyses were performed using SPSS 16.0 for Windows (SPSS Inc., Chicago, IL, USA ). Continuous variables were analyzed for normality of the distribution using the Kolmogorov-Smirnov test (One-Sample Kolmogorov-Smirnov D-Test). The continuous variables with normal distribution were compared between two or more independent groups by Student *t*-test or One-way ANOVA test with LSD Post hoc analysis, while those variables with non-normal distribution were compared by using Mann-Whitney U test or Kruskal-Wallis H test, respectively. The correlations between the continuous variables were assessed using the Pearson or Spearman correlation tests according to the type of the variables’ distribution. Factors with *p* < 0.05 were considered statistically significant.

## 3. Results

After analysis of the data, higher levels of Cu (*p* = 0.002) and Zn were found in the control group compared to COPD patients, but as for Zn there was no statistical significance (*p* = 0.397). A trend was observed for the concentration of Pb (*p* = 0.747) and Cd (*p* = 0.184) to be higher in COPD patients compared to controls, although the difference was still not statistically significant. Probably, if a larger number of patients were studied, this difference would be clearly noted (Table 2).

In controls a high degree of negative correlation between age and Pb concentration in serum (r = −0.927, *p* = 0.023) was found. In the female control group, a statistically significant negative correlation was observed between age and serum Zn level (r = −0.962; *p* = 0.0009). For Pb, this tendency was not statistically proven (r = −0.771, *p* = 0.072).

In COPD patients, we found a positive correlation between BMI and serum Cu levels (r = 0.413, *p* = 0.005). When splitting the patients by sex, a higher concentration of serum Cu was found in men with BMI ≥ 30 compared to those with BMI < 30 (359.16 ± 78.97 μg/L vs. 204.76 ± 338.36 μg/L, *p* = 0.044). A tendency for higher Cu serum levels was found in non-common exacerbators compared to common exacerbators (265.7164 ± 42.4912 μg/L vs. 205.0667 ± 78.3289 μg/L, *p* = 0.089). Interestingly, in women patients this was on the contrary—common exacerbates had higher serum Cu compared to non-common (3.0533 ± 5.71 μg/L vs. 1.9431 ± 6.66 μg/L, *p* = 0.059).

In the group of COPD, a positive correlation between Cd and Pb in the serum was present (r = 0.758, *p* = 0.0001). Cd was probably higher in asymptomatic patients than in symptomatic patients (0.4710 ± 0.0611 μg/L vs. 0.3848 ± 0.1146 μg/L, *p* = 0.061), but in the specific number of patients studied, the difference was statistically unproven. Near to significance higher concentration of Cd and significantly higher Pb in the serum of COPD patients without therapy with inhaled corticosteroids (ICS) compared to those with ICS was found (0.4330 ± 0.1135 μg/L vs. 0.3687 ± 0.1017 μg/L, *p* = 0.054 for Cd, and 0.0811 ± 0.0166 μg/L vs. 0.0707 ± 0.0129 μg/L, *p* = 0.025 for Pb). Cadmium correlated positively with Zn (r = 0.424, *p* = 0.003) and Pb (r = 0.758, *p* = 0.003). In patients, not very strong but significant positive correlation was found between Cd and C-reactive protein (CRP) (r = 0.380, *p* = 0.019, Figure 1) and Pb and CRP (r = 0.452, *p* = 0.004, Figure 2). Patients with high CRP had significantly higher levels of Pb in the serum compared to those with CRP in the referent values (0.0811 ± 0.0152 μg/L vs. 0.0711 ± 0.0131 μg/L, *p* = 0.035). A positive correlation of Pb with Zn (r = 0.517, *p* < 0.001) and visfatin (r = 0.436, *p* = 0.006) was found in COPD patients.

Patients with stable or decreased forced expiratory volume in 1 s (FEV_1_)/(FVC) ratio tended to have higher Pb concentration in the serum compared to those with increased FEV_1_/FVC (0.0803 ± 0.01837 μg/L vs. 0.0715 ± 0.0117 μg/L, *p* = 0.067). A tendency for higher Pb was also found in COPD patients with low hemoglobin levels compared to those with normal hemoglobin (0.0807 ± 0.0116 μg/L vs. 0.0719 ± 0.0161 μg/L, *p* = 0.067). In addition, not very strong but significant negative correlation between Pb in the serum and hemoglobin was demonstrated (r = −0.356, *p* = 0.028, Figure 3).

## 4. Discussion

The importance of the heavy metals copper and zinc in COPD patients and control individuals should be interpreted in terms of their biochemical role in the body. Cu is a biogenic element, which is involved in the structure of the antioxidant enzymes catalase, peroxidase, etc., which hinder the formation of ROS, and in cytochrome oxidase—a terminal enzyme of the respiratory chain in mitochondria—and is therefore necessary for the energy generation processes in cells [29].

The concentration of free copper and cupro ion (Cu^2+^) increases in direct proportion to the acidity of the water. Copper hydroxide predominates in waters with a pH of 8 or higher [31].

The cellular toxicity of the heavy metal copper can be explained by its involvement in the Fenton reaction. The cupro ion (Cu^+^) is able to catalyze the formation of hydroxyl radicals. The oxidative stress leads to subsequent destruction of lung tissue, anemia, kidney and liver disease [29,32].

A similar trend, but at a smaller percentage, is observed in the trace element Zn, which is known to be useful for the body in certain quantities. Zinc is involved in the structure of Carbonic anhydrase B, Carboxypeptidase A, Glutamate dehydrogenase and other enzyme systems [33]. This metal is also part of the hormone insulin, which is the only hormone that lowers blood glucose concentration [34].

Dietary Zn is a beneficial and nutritionally essential element. Zinc has been shown in animal studies to be antagonistic to cadmium tumorigenicity [35]. In humans, reduced zinc concentrations are associated with adverse health outcomes including impaired immune system function [36].

From a biochemical point of view, zinc ions (Zn^2+^) play a key role in many processes, such as DNA and RNA synthesis, energy metabolism, many metabolic reactions and in the regulation of the immune response [29,35].

Increased prevalence of obstructive pulmonary disease has been associated with low daily dietary Zn intake [15]. This is thought to be partially due to the protective effects of Zn against cadmium, whose metal ion has similar chemistry but high toxicity and accumulates in large doses in smokers [37,38].

As it has been shown in other studies, Cu and Zn possess synergistic action in respect of the total antioxidant activity by preventing the formation of ROS. These metals are involved in the first and second levels of antioxidant protection of the cell, preventing the peroxidation of lipids, proteins and nucleic acids. In a number of studies [39,40], elevated levels of the trace elements Cu and Zn have been found in the control group compared to patients with COPD. Many trace elements play important roles in the activation or inhibition of enzymatic reactions or the oxidant/antioxidant balance. Hence, trace elements may be involved either directly or indirectly in the pathogenesis of several diseases, including COPD.

One constituent of tobacco and tobacco smoke, Cd, is of considerable interest because it is toxic and because tobacco use is a major environmental source of Cd exposure in the general population [17,19,27]. A potential mechanism of Cd co-carcinogenicity that involves competitive Zn displacement by Cd from several DNA repair enzymes has been described [41]. In addition to inhibiting DNA repair enzymes, Cd exposure causes an inflammatory response by stimulating reactive oxygen species production by human polymorphonuclear leukocytes and phagocytic cells. This highly toxic heavy metal affects antioxidant enzymes, especially superoxide dismutase and catalase, and is able to replace Cu and Fe in some proteins [42]. It is also possible to directly affect the function of binding proteins, leading to disruption of barrier function, disorganizing the cross-linking of collagen and elastin, stabilizing the extracellular matrix and accelerating collagen and elastin damage leading to emphysema [43,44,45,46,47].

Lead is characterized by multilateral toxicity. This metal has low chemical mobility and large distribution. It exists in the form of metallic lead, inorganic ions and salts [9]. It is found in airborne dust particles and enters the aquatic environment through ore mining, the production of batteries, chemicals and dyes.

Lead not only does not play a significant role from a biochemical aspect, but has a powerful toxic effect on all organisms. Over 90% of Pb enters erythrocytes. At high doses, this metal exhibits embryo toxic effects [26], causes disorders of hematopoiesis and erythropoiesis [25], neurotoxic effects, and binding to thiol groups [45].

Lead exposure occurs mainly via the food chain and water consumption, which is the main route of exposure for the non-smoking adult population. Exposure to Pb by inhalation in the form of dust or vapor depends on several factors, such as occupation, tobacco use and leisure activities [12,48].

Because cadmium and lead are heavy metals and toxicants, the mechanism of biological action is similar in both elements. It is based on the ability of these metals to inhibit enzymatic and hormonal processes that are strategically important for the body’s metabolism.

Our studies showed that male individuals with COPD and BMI ≥ 30 have higher levels of biogenic element Cu compared to COPD male’s with BMI < 30. The increased BMI index is associated with a slow overall metabolism of individuals [49], which largely explains the results obtained. This result can be explained by the fact that at low speed the exchange is reduced, as well as by the body’s ability to score this element in the composition of necessary enzymes. In this aspect, Huber et al. [49] in their study found that weight loss remains an important strategy that can improve the health of patients with COPD in mild to severe disease.

The powerful toxic effect in somatic and sex cells of Cd can be reduced from the metals Zn, cobalt, selenium and thiol compounds [50,51,52,53,54,55,56]. The mechanisms by which these compounds alleviate Cd toxicity have not yet been clarified. For example, the protective action of Zn does not depend on de-creased absorption of Cd in tissues [57,58,59]. Zinc and Cd have been known as antagonists [60,61]. The toxicity of Cd, which has similarities to Zn deficiency, can be reduced with the addition of Zn [62,63]. 

Many interesting results of the correlation between Cd and Pb with CRP were found. A significant degree of reliability between the concentration of these metals and this acute phase protein indicates that Pb and Cd may be related to the inflammatory process, as previously described [64,65]. According to our results, it seems that these heavy metals are involved in the exacerbation of COPD symptoms, probably along with the formation of ROS.

From a biochemical point of view, the correlations of the heavy metals Pb and Cd with Zn are also intriguing. They probably participate in similar biochemical complexes and reactions. These interferences may be due to the similar chemical properties of these metals and their similar mechanism of participation in metabolism [66].

## 5. Conclusions

In patients with COPD, the levels of Cu (*p* = 0.002) and Zn were lower compared to the control group; as for Zn, there was no statistical significance (*p* = 0.397). This is probably because these elements are biogenic and are used in the body for its antioxidant protection. In fact, it cannot be stated with certainty that elevated levels of Cu and Zn in the environment have a negative impact in COPD patients. There was a trend towards higher levels of toxicants lead and cadmium in the COPD patients compared to the control group of patients. There is a statistically unproven trend toward higher levels of lead and cadmium in COPD patients compared to controls, which to some extent supports our hypothesis that there is a relationship between environmental lead and cadmium levels and COPD manifest. In COPD patients, a positive correlation was found between BMI and serum Cu levels (r = 0.413, *p* = 0.005). A higher concentration of serum Cu was found in men with BMI ≥ 30, compared to those with BMI < 30. In COPD patients, a tendency for higher Cu serum levels was found in non-common exacerbators compared to common exacerbators. In women patients, this was opposite—common exacerbates had higher serum Cu compared to non-common. In the group of COPD, a positive correlation between Cd and Pb in the serum was present (r = 0.758, *p* = 0.0001). A significant positive correlation was observed between the element zinc and the toxic metals lead (*p* = 0.003) and cadmium (*p* = 0.003). There is also a positive correlation to a lesser extent between CRP and cadmium (r = 0.380; *p* = 0.019) and lead (r = 0.452; *p* = 0.004). The correlation of lead and cadmium with CRP also shows that these elements may also be associated with the presence of inflammatory processes. A significant negative correlation exists between Pb in the serum of patients with COPD and their blood hemoglobin (r = −356; *p* = 0.028). The results of our study suggest that higher doses of the trace elements Cu and Zn do not always have a negative effect in patients with COPD, while the toxicants Pb and Cd may be involved in COPD exacerbation and can be used as prognostic biomarkers for progression. Further studies are warranted to confirm these preliminary results.

## Figures and Tables

**Figure 1 ijerph-20-01427-f001:**
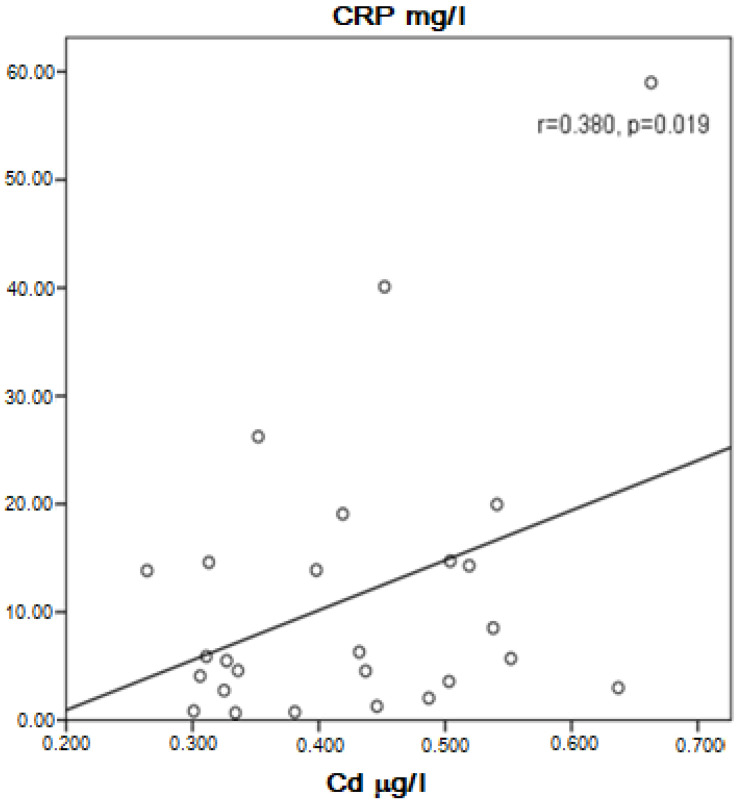
Correlation of Cd and CRP in COPD patients.

**Figure 2 ijerph-20-01427-f002:**
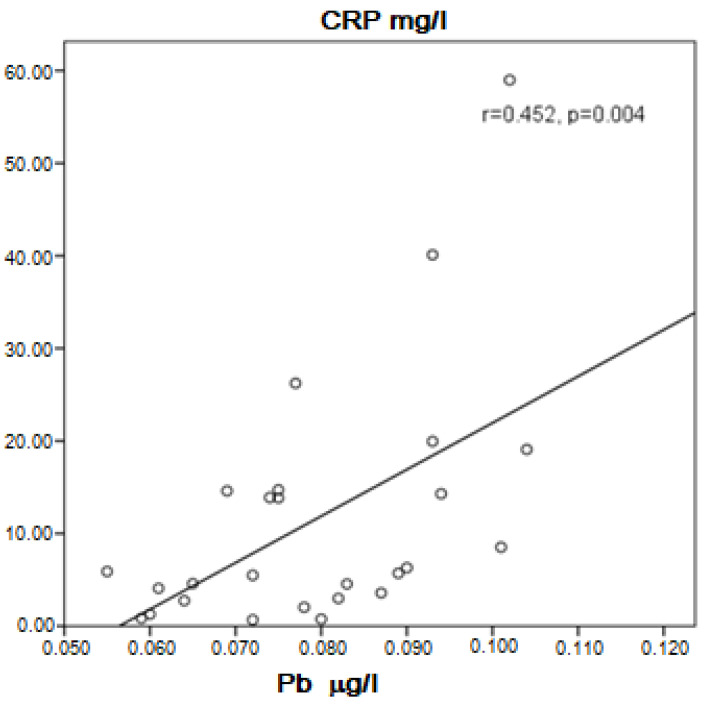
Correlation of Pb and CRP in COPD patients.

**Figure 3 ijerph-20-01427-f003:**
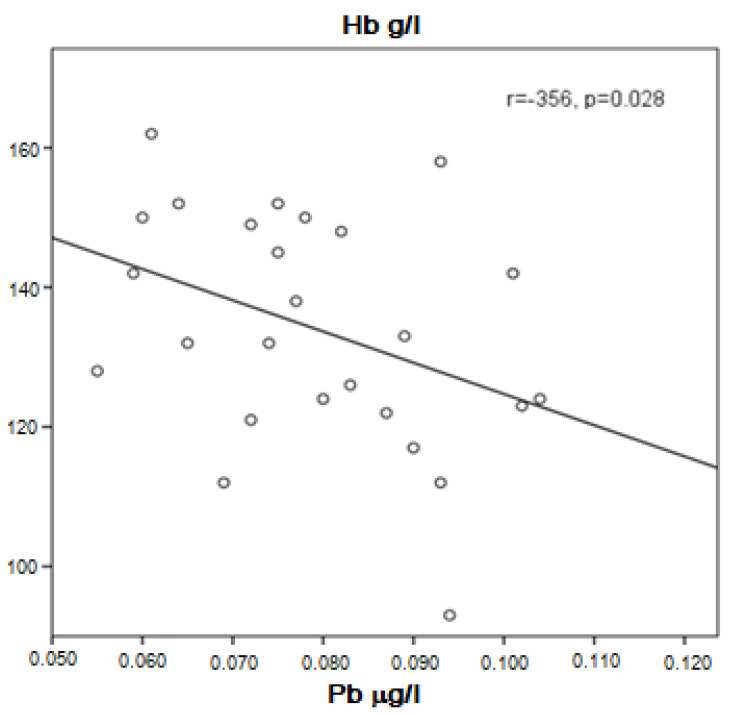
Correlation of Pb and hemoglobin in COPD patients.

**Table 1 ijerph-20-01427-t001:** Demographic and clinical data of COPD patients and controls.

Characteristics	COPD Patients	Controls
Number	(*n* = 54)	(*n* = 22)
males	33 (75%)	11 (50.0%)
females	21 (25%)	11 (50.0%)
Age at the inclusion in the study		
mean ± SD (years)	68 ± 6.73	54 ± 19.7
median (range) (years)	68.5 (50–79)	48 (34–84)
BMI (body mass index)	(*n* = 54)	(*n* = 22)
normal (18.5–24.9)	12 (22.2%)	7 (33.3%)
overweight (25–29.9)	24 (44.4%)	15 (66.7%)
obese (≥30)	18 (33.4%)	0 (0%)
Smoking status	(*n* = 54)	(*n* = 22)
non—smokers	5 (9.1%)	14 (66.6%)
ex-smokers	35 (65.9%)	4 (16.7%)
current smokers	14 (25%)	4 (16.7%)
Smoking habits (packs/year)		
mean ± SD (years)	36.1 ± 16.5	32.5 ± 38.9
median (range)	35 (15–90)	32.5 (5–60)
Occupational hazard		
non present	42 (77.3%)	
present	12 (22.7%)	
COPD stage	(*n* = 54)	
GOLD I	2 (4.5%)	
GOLD II	34 (63.6%)	
GOLD III	16 (29.5%)	
GOLD IV	2 (2.3%)	
CAT test scores		
mean ± SD,	12.47 ± 3.71	
median (range)	12.5 (7–21)	
FEV1% pred.		
mean ± SD	54.4 ± 16.11	
FEV1/FVC%		
mean ± SD	70.95 ± 15.35	

**Table 2 ijerph-20-01427-t002:** Serum concentration of trace elements in controls and COPD patients.

Trace Element	Controls	COPD	*p*
Cu	374.29 ±15.03 μg/L	238.55 ± 175.31 μg/L	*p* = 0.002
Zn	2010.44 ± 670.01 μg/L	1672.78 ± 934.27 μg/L	*p* = 0.397
Pb	0.0732 ± 0. 009 μg/L	0.075 ± 0.0153 μg/L	*p* = 0.747
Cd	0.334 ± 0.0216 μg/L	0.395 ± 0.110 μg/L	*p* = 0.184

## Data Availability

Not applicable.

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
