# Peer review of "The Serum Levels of the Heavy Metals Cu, Zn, Cd, and Pb and Progression of COPD—A Preliminary Study"

_ijerph, 2023, doi:10.3390/ijerph20021427_

Round 1
Reviewer 1 Report
This study assessed the serum levels of Cu, Zn, Cd, and Pb in the serum of patients with COPD and unaffected by the disease control group and evaluated their possible role as markers in progression of the disease. This manuscript conducted a systematic and logical work. This study is an important contribution to understand the possible role for the selected metals as biomarkers for COPD. Hence, I recommend it for publication in the journal. However, this article cannot be accepted at current status until a major revision is finished. There are a number of reasons why I recommend major revision of the paper. If the authors choose to revise the article, the following questions and concerns must be addressed:
1. The full name of the abbreviation is needed where it first appeared in the text.
2. "Abstract" is suggested to be modified with a clear focus on the findings and contribution of your study.
3. The introduction is disorderly and unsystematic; it needed to be greatly improved. The authors should rearrange and highly summary the newly relevant studies, the references should update to recent years. And, the objectives of this text should be clearly stated.
4. The biggest problem of the text is the section of discussion. In total, the discussion is superficial, and some conclusions and viewpoints are unconvincing or lack of evidences or references, such as “Because cadmium and lead are heavy metals and toxicants, the mechanism of bio-236 logical action is similar in both elements”. In addition, some discussions are not relevant to the study topic of this study; in fact, some of them are belong to “introduction statement” and should be moved to the section of Introduction. The authors should dig the data deeply and draw conclusions based on previous studies.
Author Response
Questions and suggestions of Reviewer 1 and Response of authors:
- The full name of the abbreviation is needed where it first appeared in the text.
Response: The full names of abbreviations have been removed, excluding their first appearance.
- "Abstract" is suggested to be modified with a clear focus on the findings and contribution of your study.
Response: Thank you for the revision. The abstract was modified with a clear focus on the findings and contribution of our study.
- The introduction is disorderly and unsystematic; it needed to be greatly improved. The authors should rearrange and highly summary the newly relevant studies, the references should update to recent years. And, the objectives of this text should be clearly stated.
Response: Thank you for the revision. The introduction has been improved. The objectives of the text are clearly stated.
- The biggest problem of the text is the section of discussion. In total, the discussion is superficial, and some conclusions and viewpoints are unconvincing or lack of evidences or references, such as “Because cadmium and lead are heavy metals and toxicants, the mechanism of bio-236 logical action is similar in both elements”. In addition, some discussions are not relevant to the study topic of this study; in fact, some of them are belong to “introduction statement” and should be moved to the section of Introduction. The authors should dig the data deeply and draw conclusions based on previous studies.
Response: Thank you for the revision. Discussion has been revised. Some parts of it have been moved to the "Introduction" section.
Reviewer 2 Report
The paper presented for review “The serum levels of the trace element Cu, Zn, Cd, and Pb and 2 progression of COPD – a preliminary study” is interesting it should be published in the IJERPH journal. However, it has to be corrected prior to publication. Problems were observed in the writing of the text and the lack of references in the information provided. Following my suggestions:
Abstract
Line 12-15: I recommend rewriting this sentence as it is a bit confusing. Here's my suggestion: In this preliminary study, we used atomic absorption spectroscopy to measure the levels of 4 trace elements (Cu, Zn, Cd, and Pb) in the blood (serum) of patients with COPD and a control group without the disease. We examined whether these elements might serve as markers for the progression of COPD.
Line 17:I recommend using 'control group' instead of 'unaffected by the disease control group'
Line 31: I recommend using keywords that have not been used in the title.
Introduction
Sentences from lines 41-42, and 42-44 are missing a reference at the end.
Lines 55 -59: Rewrite recommendation: The region where the settlement subjected to research is located is also significant. The Stara Zagora region in Bulgaria is one of the risk regions in terms of environmental pollution, due to the presence of three large coal-fired thermal power plants in the Maritsa-Iztok complex nearby (Reference).
Line 61: Please, provide the reference of this sentence.
Line 65: Please connect those sentences “Contained in airborne dust particles. It enters the..”, and please provide the reference of this sentence.
Line 89: Please, provide the reference of this sentence.
In the last paragraph, there should be a hypothesis formed, which will then be developed (tested) in the following sections.
Line 95: That sentence was not clear: "In the present study a total of 50 participants from the region", because in table 1 a total of 76 participants were shown.
Line 114: Missed putting the phrase "Determination of quantities of heavy metals Cd, Pb, Cu and Zn" in italics, as was done in other topics.
Topic Determination of quantities of heavy metals Cd, Pb, Cu, and Zn: Please give more details, such as the brand and grade of reagents used.
Results
Lines 133-134: If there was no statistically significant difference, it cannot be accurately stated that the concentration of Pb or Cd was higher in one group compared to the other. The concentration of Pb (p=0.747) and Cd (p=0.184) was not significantly different between patients with COPD and controls. This accurately reflects the results of the statistical test and avoids making a statement that is not supported by the data. It is important to carefully interpret and present statistical results in order to accurately convey the findings of a study.
Lines 137-138: Please rewrite the sentence more clearly.
On line 159 the authors said: "Factors with p<0.05 were considered statistically significant". And in Lines 148-149 the p=0.061 was considered significant. Please explain this fact better.
Line 220: Please correct the reference that is out of standard
Conclusion
First, there should be a summary of the tested hypotheses - have they been confirmed? Then, the results should be presented in a global context. How does this research compare to what was already known from other studies on this topic?
The manuscript should be more than a report on conducted measurements. It should be placed within the context of the existing knowledge (published works).
Author Response
Questions and suggestions of Reviewer 2 and Response of authors:
Abstract
Line 12-15: I recommend rewriting this sentence as it is a bit confusing. Here's my suggestion: In this preliminary study, we used atomic absorption spectroscopy to measure the levels of 4 trace elements (Cu, Zn, Cd, and Pb) in the blood (serum) of patients with COPD and a control group without the disease. We examined whether these elements might serve as markers for the progression of COPD.
Response: Thank you for the suggestion. The sentence has been rewritten taking your suggestion into account.
Line 17: I recommend using 'control group' instead of 'unaffected by the disease control group'
Response: Thanks for the recommendation. In the text, we have replaced the phrases “unaffected by the disease control group” with “control group”.
Line 31: I recommend using keywords that have not been used in the title.
Response: Thank you for the recommendation. We replaced some of the keywords with ones not used in the title.
Introduction
Sentences from lines 41-42, and 42-44 are missing a reference at the end.
Response: References Have been added at the end to the sentences from lines 41-42 and 42-44.
Lines 55 -59: Rewrite recommendation: The region where the settlement subjected to research is located is also significant. The Stara Zagora region in Bulgaria is one of the risk regions in terms of environmental pollution, due to the presence of three large coal-fired thermal power plants in the Maritsa-Iztok complex nearby (Reference).
Response: Thanks for the recommendation. The text on lines 55 to 59 has been rewritten to accommodate your suggestion. Also added a reference at the end.
Line 61: Please, provide the reference of this sentence.
Response: A reference was added at the end.
Line 65: Please connect those sentences “Contained in airborne dust particles. It enters the..”, and please provide the reference of this sentence.
Response: The sentences on line 65 were connected. An end of sentence reference was also added.
Line 89: Please, provide the reference of this sentence.
Response: A reference was added at the end of the sentence.
In the last paragraph, there should be a hypothesis formed, which will then be developed (tested) in the following sections.
Response: In the last paragraph, a hypothesis was formed, which was then developed (tested) in the following sections.
Line 95: That sentence was not clear: "In the present study a total of 50 participants from the region", because in table 1 a total of 76 participants were shown.
Response: The sentence on line 95 has been corrected to the actual number of patients (76).
Line 114: Missed putting the phrase "Determination of quantities of heavy metals Cd, Pb, Cu and Zn" in italics, as was done in other topics.
Response: The phrase "Determination of the amounts of heavy metals Cd, Pb, Cu, and Zn" has been made in italics.
Topic Determination of quantities of heavy metals Cd, Pb, Cu, and Zn: Please give more details, such as the brand and grade of reagents used.
Response: Added information regarding the brand and grade of reagents used to the topic "Determination of heavy metals Cd, Pb, Cu and Zn".
Results
Lines 133-134: If there was no statistically significant difference, it cannot be accurately stated that the concentration of Pb or Cd was higher in one group compared to the other. The concentration of Pb (p=0.747) and Cd (p=0.184) was not significantly different between patients with COPD and controls. This accurately reflects the results of the statistical test and avoids making a statement that is not supported by the data. It is important to carefully interpret and present statistical results in order to accurately convey the findings of a study.
Response: The information on lines 133 and 134 has been edited and conformed to your recommendations.
Lines 137-138: Please rewrite the sentence more clearly.
Response: The sentence at lines 137 and 138 was rewritten.
On line 159 the authors said: "Factors with p<0.05 were considered statistically significant". And in Lines 148-149 the p=0.061 was considered significant. Please explain this fact better.
Response: Corrected the information on lines 148 and 149 about the degree of confidence of these results.
Line 220: Please correct the reference that is out of standard
Response: On line 220, the necessary correction has been made.
Conclusion
First, there should be a summary of the tested hypotheses - have they been confirmed? Then, the results should be presented in a global context. How does this research compare to what was already known from other studies on this topic?
Response: The conclusion has been revised, with the text aligned with your recommendations.
The manuscript should be more than a report on conducted measurements. It should be placed within the context of the existing knowledge (published works).
Response: A thorough revision of the manuscript has been made based on your recommendations.
Round 2
Reviewer 2 Report
Based on my review of the paper, I believe it is ready for publication. However, I have identified a few minor details that should be corrected before it is published. Overall, the paper demonstrates a significant improvement and I recommend moving forward with its publication. Abstract: Line 22: Please change 'sup-ports' to 'supports' Line 23: Please change 'cad-mium' to 'cadmium' Line 28: Please change 'pres-ence' to 'presence' In both the abstract and conclusion, I feel that there is a lack of a concluding sentence that summarizes the main ideas and offers future recommendations. Both sections only end with a description of the results. Thank you!
Author Response
Questions and suggestions of Reviewer 2 of Round 2 and Response of authors:
Abstract: Line 22: Please change 'sup-ports' to 'supports' Line 23: Please change 'cad-mium' to 'cadmium' Line 28: Please change 'pres-ence' to 'presence'.
Response: The text of lines 22, 23 and 28 has been corrected.
In both the abstract and conclusion, I feel that there is a lack of a concluding sentence that summarizes the main ideas and offers future recommendations. Both sections only end with a description of the results.
Response: Concluding sentences have been added to summarize the main ideas and to suggest future recommendations.